# Effect of grey-level discretization on texture feature on different weighted MRI images of diverse disease groups

Gergő Veres[1]*, Norman Félix Vas[2], Martin Lyngby Lassen[3,4], Monika Béresová[1], Aron K. Krizsan[5], Attila Forgács[5], Ervin Berényi[1], László Balkay[2]

1 Division of Radiology and Imaging Science, Department of Medical Imaging, Faculty of Medicine, University of Debrecen, Debrecen, Hungary, 2 Division of Nuclear Medicine and Translational Imaging, Department of Medical Imaging, Faculty of Medicine, University of Debrecen, Debrecen, Hungary, 3 Cedars-Sinai Medical Center, AIM Group, Los Angeles, CA, United States of America, 4 Department of Clinical Physiology, Nuclear Medicine and PET and Cluster for Molecular Imaging, University of Copenhagen, Copenhagen, Denmark, 5 Scanomed Nuclear Medicine Center, Debrecen, Hungary

* veres.gergo@med.unideb.hu

**Data Availability Statement:** All relevant data are within the paper and its Supporting Information files.

## Abstract

### Purpose

Many studies of MRI radiomics do not include the discretization method used for the analyses, which might indicate that the discretization methods used are considered irrelevant. Our goals were to compare three frequently used discretization methods (lesion relative resampling (LRR), lesion absolute resampling (LAR) and absolute resampling (AR)) applied to the same data set, along with two different lesion segmentation approaches.

### Methods

We analyzed the effects of altering bin widths or bin numbers for the three different sampling methods using 40 texture indices (TIs). The impact was evaluated on brain MRI studies obtained for 71 patients divided into three different disease groups: multiple sclerosis (MS, N = 22), ischemic stroke (IS, N = 22), cancer patients (N = 27). Two different MRI acquisition protocols were considered for all patients, a T2- and a post-contrast 3D T1-weighted MRI sequence. Elliptical and manually drawn VOIs were employed for both imaging series. Three different types of gray-level discretization methods were used: LRR, LAR and AR. Hypothesis tests were done among all diseased and control areas to compare the TI values in these areas. We also did correlation analyses between TI values and lesion volumes.

### Results

In general, no significant differences were reported in the results when employing the AR and LAR discretization methods. It was found that employing 38 TIs introduced variation in the results when the number of bin parameters was altered, suggesting that both the degree and direction of monotonicity between each TI value and binning parameters were characteristic for each TI. Furthermore, while TIs were changing with altering binning values, no changes correlated to neither disease nor the MRI sequence. We found that most indices

**Funding:** The research was supported by the Thematic Excellence Programme (TKP2020-NKA-04) of the Ministry for Innovation and Technology in Hungary. The funders had no role in study design, data collection and analysis, decision to publish, or preparation of the manuscript.

**Competing interests:** The authors have declared that no competing interests exist.

correlated weakly with the volume, while the correlation coefficients were independent of both diseases analyzed and MR contrast. Several cooccurrence-matrix based texture parameters show a definite higher correlation when employing the LRR discretization method However, with the best correlations obtained for the manually drawn VOI. Hypothesis tests among all disease and control areas (co-lateral hemisphere) revealed that the AR or LAR discretization techniques provide more suitable texture features than LRR. In addition, the manually drawn segmentation gave fewer significantly different TIs than the ellipsoid segmentations. In addition, the amount of TIs with significant differences was increasing with increasing the number of bins, or decreasing bin widths.

## Conclusion

Our findings indicate that the AR discretization method may offer the best texture analysis in MR image assessments. Employing too many bins or too large bin widths might reduce the selection of TIs that can be used for differential diagnosis. In general, more statistically different TIs were observed for elliptical segmentations when compared to the manually drawn VOIs. In the texture analysis of MR studies, studies and publications should report on all important parameters and methods related to data collection, corrections, normalization, discretization, and segmentation.

## Introduction

Magnetic resonance imaging (MRI) is widely used in neurology to obtain differential diagnoses, characterize and confirm multiple types of lesions, tumor staging and therapy planning, and evaluate therapy response [1–6]. Using MRI, various anatomical lesions, pathological progressing, and functional changes can be characterized based on visual appearance and underlying texture analyses [7–11]. However, quantitative measures have been shown to improve the diagnostic outcome for patients when compared to standard qualitative assessments. In the clinical setting, the lesion volume is the most common quantitative measurement used to assess neurological disorders; however, there is a growing body of interest to employ radiomics features for quantitative assessments (intensity, texture, or shape). Previous studies have shown promising results when employing texture (heterogeneity), hereunder that texture features might outperform conventional measures (VOI size and mean value) and visual inspection [12–18]. Despite promising results, the use of texture indices is sensitive to factors such as acquisition protocols and physical parameters of the imaging systems [19–21]. In addition, texture indices are also sensitive to the tumor segmentation method and the discretization steps [22, 23]. Regardless of the types of texture indices used, the discretization (binning or resampling) is always applied, converting the original voxel values of the volume of interest into a predefined range of values. Thus, the discretization method will inherently influence the value of the TIs tailored to characterize the texture of the lesion. The impact of image discretization has already been investigated in numerous studies in nuclear medicine and CT [24–27], however, the discretization methods and their impact has not been explored thoroughly in MRI [28, 29]. In brief, two distinct discretization methods have been proposed for MRI: relative and absolute resampling. The relative method discretizes the voxel-values into a fixed number of bins, in contrast, the absolute resampling employs fixed bin sizes. The most recent guidelines released by the Image Biomarker Standardization Initiative (IBSI) [30] recommend relative discretization techniques for MRI. Despite the recommendations, recent studies have

shown that the relative discretization method might not be the optimal technique [29, 31]. IBSI uses the abbreviations FBN and FBS for the above two methods, however, since two types of FBS discretizations exist, the following abbreviations are also used: LRR instead of FBN, and AR and LAR as two subtypes of FBS [26, 32]. Recently, Carré et al. analyzed the impact of the normalization and discretization methods on radiomics features with T1 and T2 weighted MRI images [33]. Interestingly, they proposed different discretization schemas for different texture index groups, namely, the FBS for model-based first- and second-order features, and in the case of the second-order feature-based signatures the FBN is recommended. Nevertheless, in recent studies it is not uncommon that the discretization method used remains undisclosed [34–37]. Because of these contradictory results reported in studies, it is important to investigate the role of discretization methods in the radiomic features obtained for MRI images. Moreover, radiomics parameters can also be influenced by the applied segmentation method, including manual, semi-automatic or automatic delineation of the ROI or VOI. The accuracy of the calculated radiomics data may be affected by how the external tissue areas are excluded from the selected VOI, which ensures that pathological and intact tissue areas do not mix nor overlap during the evaluation [38–40]. In this work, we aimed to investigate the effect of three different discretization methods and their impact on the parameters obtained for texture calculation in a cohort of 71 patients with clinical brain studies employing MRI. For this study, three different neurological diseases were considered: multiple sclerosis (MS), ischemic stroke (IS), and tumor (TU). Applying manually defined VOIs and in addition 3D ellipsoidal VOIs placed on the inner area of the lesions, we also studied how the pixels at the lesion boundary could modify the results of the radiomics calculations.

## Methods

### MRI scanning

This retrospective study was approved by the institutional ethics committee (Regional and Institutional Ethics Committee. University of Debrecen. Clinical Center) and did not require informed consent. In addition, all patient data were fully anonymized before we accessed for image processing and data evaluation. The study comprised 71 patients who underwent contrast-enhanced 3D T1- and T2-weighted axial scans on a 1.5 Tesla Siemens Magnetom Essenza scanner. The patients were acquired in three subgroups including patients with ischemic stroke (N = 22), multiple sclerosis (N = 22), and neurological tumors (N = 27). In the oncological group, ten patients were presented with primary tumors, while 17 had metastatic disease. All MRI were performed in 2019 at the MRI imaging facility of the Kenézy Gyula University Hospital, University of Debrecen. Both T2-weighted axial and post-contrast 3D T1-weighted axial measurements were performed according to the local standard protocols of the diseases. Representative MRI images and the corresponding scan protocols are shown in Fig 1A and Table 1, respectively.

### Image processing and statistical tests

The same anatomical regions were scanned for all patients who had imaging fields positioned parallel to the bicommissural line, which links the anterior to the posterior commissure. We used the Carimas 2.10 software for VOI definition (Turku Pet Centre (http://turkupetcentre.fi/carimas/), Turku, Finland). Segmentation of the pathological areas was performed separately for the respective disease groups. Two different segmentation methods were applied in this study, a) an elliptical VOI and b) manually drawn VOIs, drawn by expert radiologists. For both segmentation methods, two VOIs were placed, one in the pathological area and one in the corresponding center on the co-lateral side of the brain. For patients with either MS or IS,

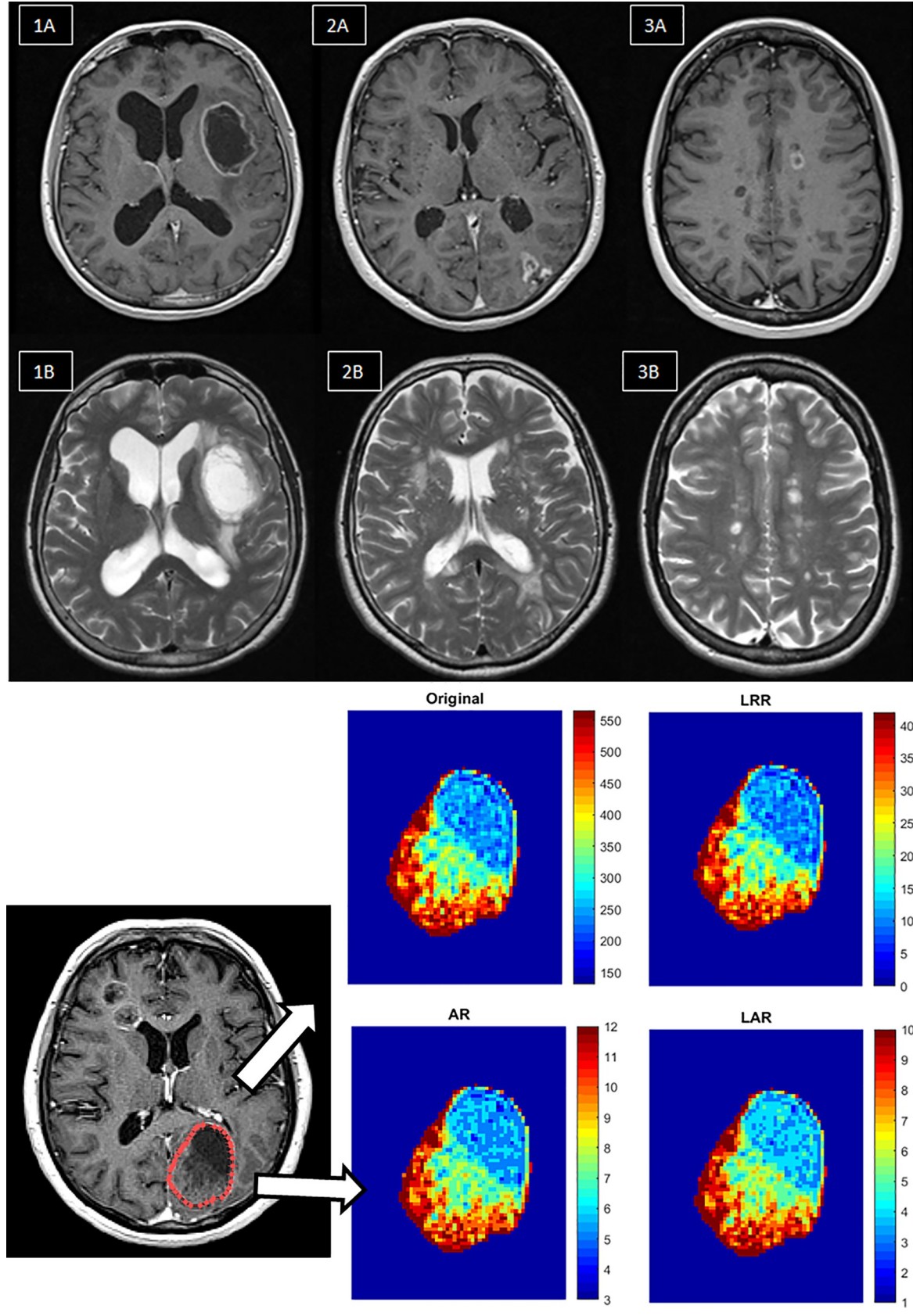

**Fig 1. a.** Representative 3D T1 post-contrast (upper row) and T2-weighted axial (lower row) MR images for the selected disease groups. 1A-1B: Glioblastoma brain tumors, 2A-2B: Ischemia, 3A-3B: MS. **b.** Visualization steps of discretization workflow in radiomics. First, images are acquired. The next step is the segmentation of the region of interest, from which texture features are extracted. Then the discretization pre-processing steps are performed on the images, namely LRR, AR and LAR. The texture characteristics are then subtracted from the area of interest. The AR and LAR discretizations smooth the contrast much better within the segment, which is most easily observed at lower values (blue color).

it is common that multiple lesions are present in one patient. To optimize feature extraction, all lesions were used for these types of diseases. For patients with IS, the areas with recent ischemia, previous vascular lesions, and healthy brain parts were considered. For these cohorts, the VOIs were enclosed using volumes of at least 1 cm$^3$. Similarly, VOIs were placed on active and inactive foci and corresponding healthy brain areas of the MS patients. The VOI related voxel data were saved by the Carimas program to a.txt file, and all further texture analysis and plotting were performed in Matlab (version 2018b, MathWorks Inc., Natick, MA). To compare the TIs between the healthy and diseased areas, non-parametric Wilcoxon rank-sum tests were also performed with the 'ranksum' Matlab function. In addition, Spearman's correlation coefficient (R) was calculated between each TI and the related lesion volumes for all diseased and healthy VOI groups and both MRI contrasts.

## Texture calculation and discretization

TIs were extracted by using the Matlab implementation of GLCM, GLSZM and GLRLM based algorithms [26], employing a total of 40 radiomics features from each VOI. The selected features were the following: 18 GLCM, 11 GLSZM, and 11 GLRLM based TIs calculated according to the IBSI guideline [30] (S1 Table in S1 File). Currently, three different methods of discretization are used for heterogeneity analyses.

The LAR method is defined by

$$I_{LAR}(i) = \left\{ \frac{I(i)}{B} \right\} - \left\{ \frac{I_{min}}{B} \right\} + 1$$

where $I_{LAR}(i)$ is the resampled intensity value of the i-th voxel, $I(i)$ is the original i-th voxel intensity, $I_{min}$ is the original minimum voxel intensity of a particular lesion, and B is the bin width. The {} brackets imply using the floor operation.

The AR method is defined by

$$I_{AR}(i) = \left[ \frac{I(i)}{B} \right] - 1$$

where $I_{AR}(i)$ is the new intensity value of the i-th voxel after applying the AR discretization. The [] brackets stand for the ceil operation. The following B values were considered: {1; 5; 10; 15; 20; 25; 30; 35; 40; 45; 50; 55; 60; 65; 70; 75; 80; 85; 90; 95; 100}. LAR and AR methods both belong to the fixed bin size IBSI definition. Since we did not intend to introduce

**Table 1. Parameters of the sequences used in this work.**

| Sequence | FOV (mm) | Slice Thickness (mm) | Gap (%) | Phase encoding direction | TR (ms) | TE (ms) | Reconstruction |
|---|---|---|---|---|---|---|---|
| T2_TSE_tra | 230 | 3 | 10 | R-L | 3760 | 86 | - |
| | Matrix size: 230x180 | | | | | | |
| 3D_T1_tra _postCM | 240 | 0,9 | 50 | R-L | 1540 | 4,73 | MPR sag/cor |
| | Matrix size: 256x256x166 | | | | | | |

R = right, L = left, TR = repetition time, TE = echo time, Sag = sagittal, Cor = coronal, MPR = multi planar reconstruction, FOV = field of view

image normalization in this work, the data was discretized to the 10th and 90th percentile of the values observed in the volume of interests [41].

Finally, the LRR method is defined by

$$I_{LRR}(i) = \left\{ \begin{array}{ll} 1 & I(i) = Imin \\ \left[ Dx \dfrac{I(i) - Imin}{Imax - Imin} \right] & otherwise \end{array} \right\}$$

where $I_{LRR}(i)$ is the new intensity value of the i-th voxel intensity after the LRR discretization, $I_{max}$ is the maximum original voxel intensity of the particular lesion and D is the number of bin parameter. We used D = {8; 16; 32; 64; 128; 256; 512; 1024}. The difference between the discretization methods are presented in Fig 1B.

## Results

In Fig 2A the representative box-and-whisker subplots show the distribution of the TIs for various discretization parameter values, and how the discretization method influences the numerical value of the investigated textural indices when employing 3D elliptical segmentations. The four presented indices presented in Fig 2A all belong to GLCM (JMax, JVar, JEntropy and Dissim); all indices are obtained from T1-weighted MRI and presented at cohort level. No differences were observed for the healthy and IS groups when comparing the bin width and bin number against TI tendencies. Furthermore, we did not find any substantial difference (>3%) between the results of AR and LAR-based discretizations or between the manual and 3D elliptical delineation methods (data are not presented).

Similar plots are presented in Fig 2B based on the T2-weighted images for 3D elliptical segmentations.

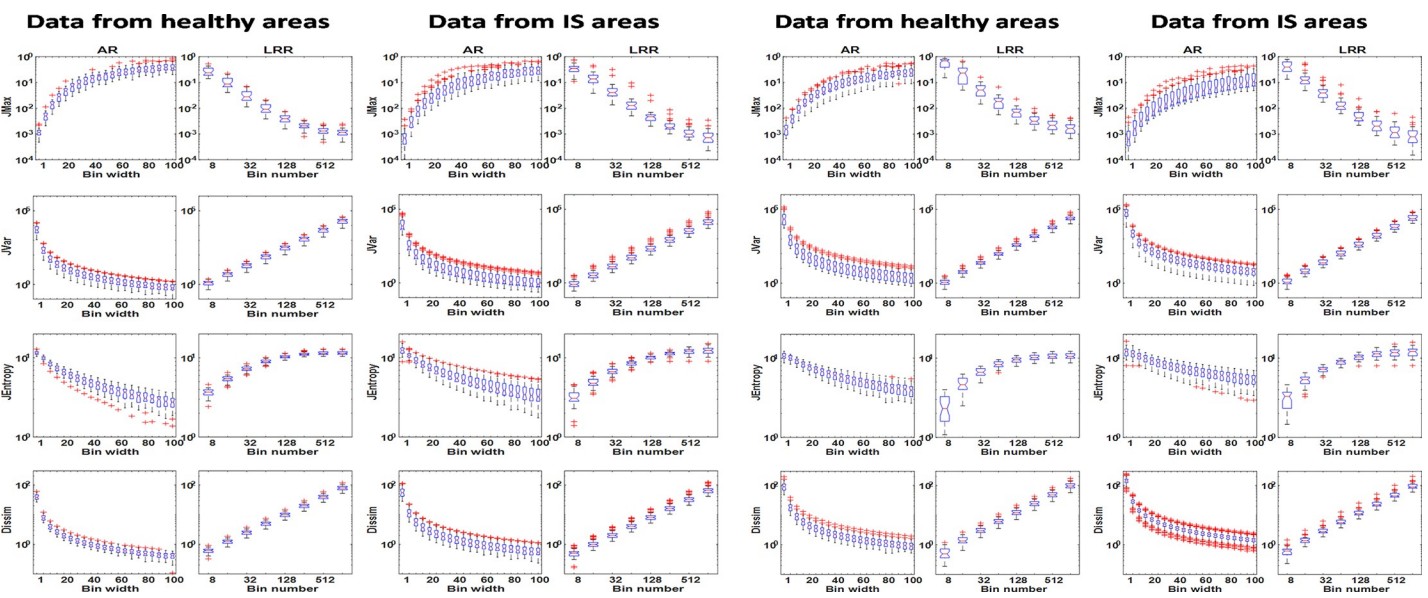

**Fig 2. a** The distribution of TIs for four selected GLCM based parameters for the IS lesions in T1-weighted images is illustrated in box plots. Each of the individual boxes represent the value of a given textural parameter extracted from the lesion dataset. The subplots are organized in a 2x2 column way, where the two columns on the left and right refer to VOIs of the healthy and pathological areas, respectively. The applied discretization parameter sets are shown on the horizontal axes and were the following: bin widths from 1 to 100 in 20 steps at the AR, and 8, 16, 32, 64, 128, 256, 512 and 1024 number of bins at the LRR method. The name of the appropriate discretization method is labeled on the top of each column. All box plots have a logarithmic scale on the y-axis. **b.** The distribution of TIs for the four selected GLCM based parameters evaluating the IS lesions and control areas in T2-weighted images is illustrated using box plots. The definitions and organizations of the charts are identical with Fig 2A.

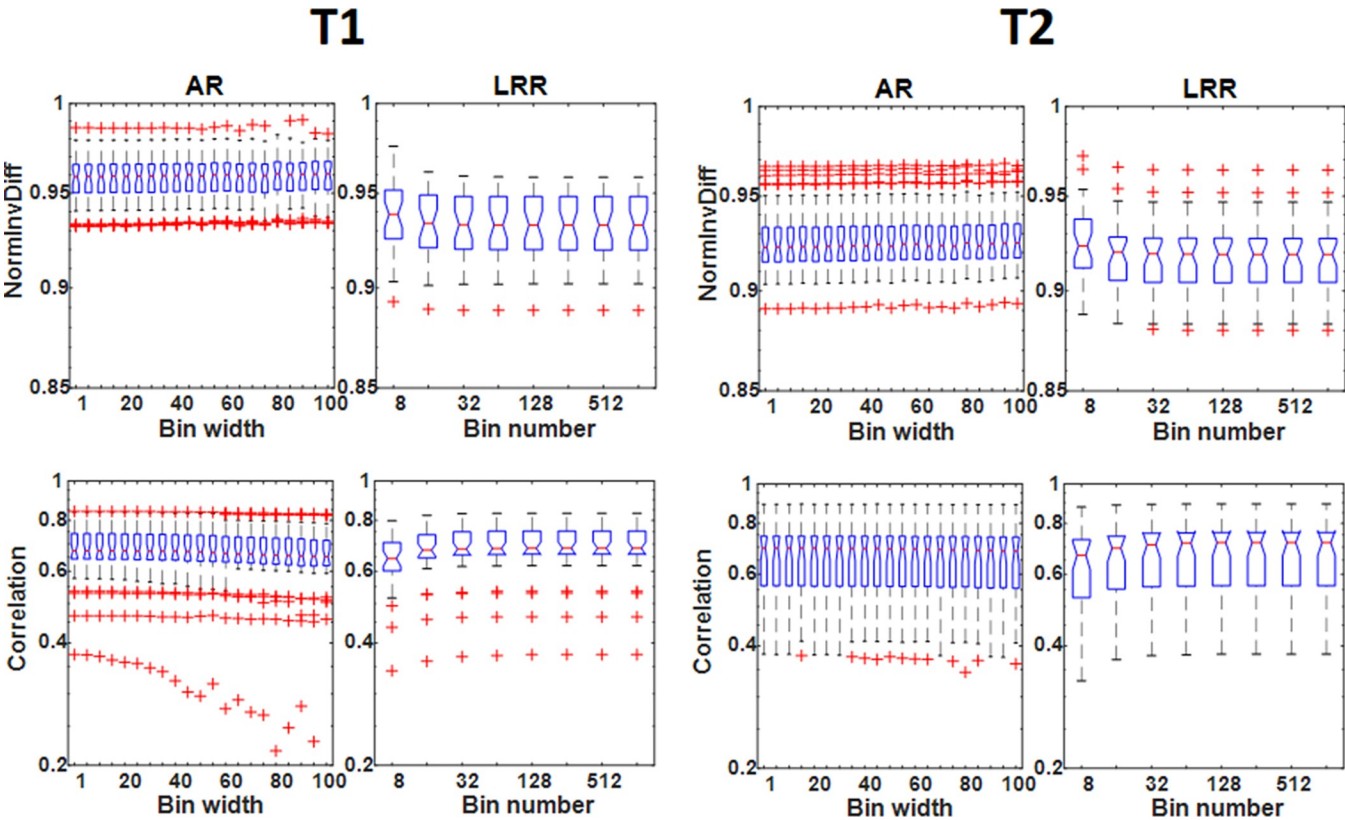

**Fig 3. The distribution of NormInvDiff and correlation GLCM type indices including all tumor VOIs is illustrated with box plots and applying elliptical VOIs.** The subplots are organized in a 4 columns way, where the two columns on the left and right refer to the T1- and T2-weighted images.

No differences in the trends obtained for the TIs for T1- and T2-weighted images were observed. Of importance, the absolute measurements were dependent on the number of bins employed. Only the parameters Correlation and NormInvDiff seem to be independent of the binning parameters as shown in Fig 3.

All 18 GLCM based TIs and the related box plots for each patient group (IS, MS, tumor) can be found in S1, S2 Figs of S1 File. Similar findings were obtained for the GLRLM and GLSZM based TIs: the T1- and the T2-weighted images do not differ in the interrelationships among the TIs and bin parameters. Representative plots of GLRLM TIs for tumor and MS areas are shown in Fig 4A and 4B respectively. All plots were based on elliptical VOI segmented data.

A similar statement can be drawn in all the other 22 TI (11 GLRLM and 11 GLSZM based TI) cases including the IS, TU, and MS groups. The corresponding 264 box plots are presented in S1 File (S3-S6 Figs). We also analyzed the correlation between all TIs and the lesion volumes, since a smaller degree of correlation is more advantageous for proper texture analysis. Altogether 480 correlation plots and the related Spearman's correlation coefficients ($R^2$) were calculated. Fig 5 shows some representative correlation scatter plots for the 3D elliptical VOIs. All 480 correlation plots can be found in S7-S12 Figs of S2 File. The related $R^2$ measures are shown in the form of color-coded maps in Fig 6A and in 6B, the earlier corresponding to elliptical VOIs and the latter corresponding to manual segmentation. The figures suggest that several parameters have a higher correlation to the segmentation volume when LRR discretization and manual segmentation are applied.

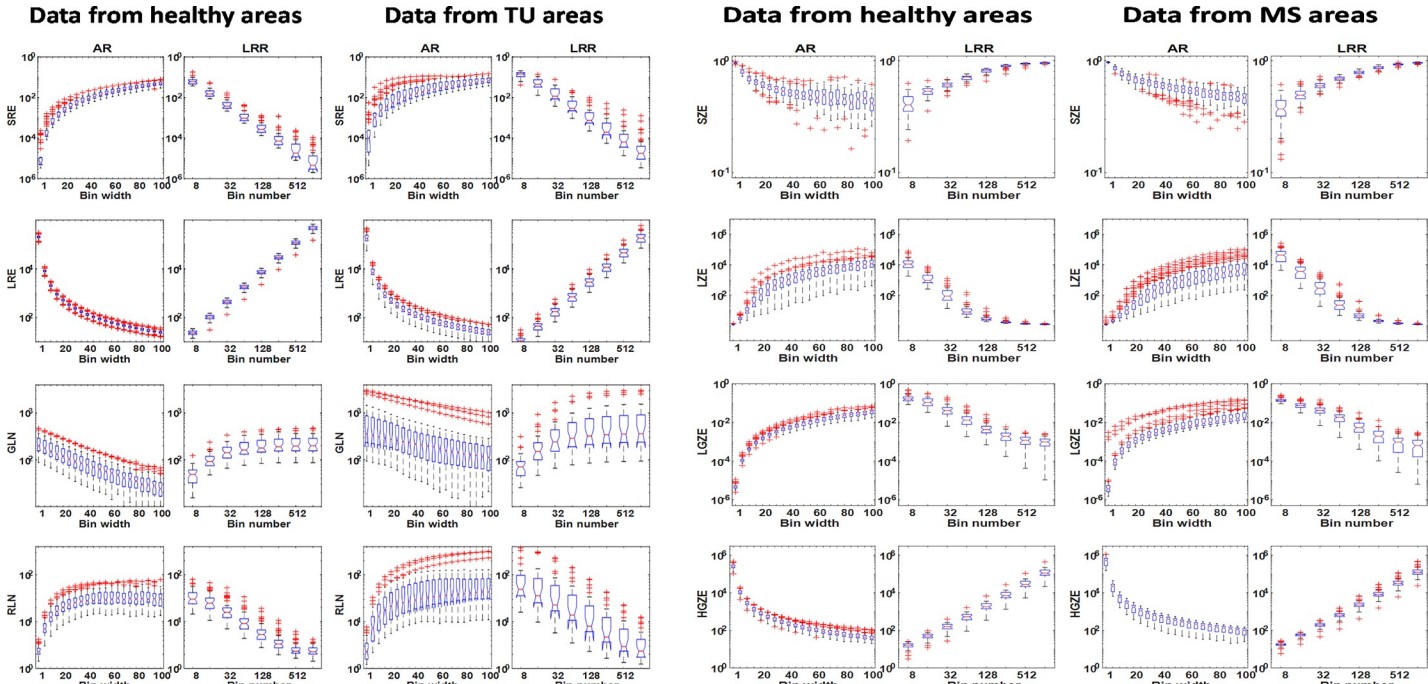

**Fig 4. a.** The distribution of TIs for four selected GLRLM based parameters in case of tumor and the control lesions at T1-weighted images. The definitions and arrangement of the charts are identical to Fig 2A. **b.** The distribution of TIs for selected four GLSZM based parameters for MS and the control lesions in the case of T2-weighted images.

As a final step, we performed Wilcoxon rank-sum hypothesis tests among all disease and control areas (Figs 7 and 8). For these calculations, we used data obtained with both segmentation methods. The assessments were obtained using fixed bin widths (B) and number of bin (D) values to 50 and 64, respectively. AR and LRR had 192 and 158 textural parameters with significant differences ($p < 0.05$) when employing elliptical segmentations, respectively, while the corresponding numbers were 179 and 119 for the manual segmentation. For higher D and/or lower B values, the number of TIs giving significant difference decreased slightly. At D = 1024 and B = 1, the number of texture parameters showing significant difference ($p < 0.05$) was 176 and 145 for the AR and LRR methods, respectively (S2 File), meaning that fewer TIs could be used in differential diagnosis when using these binning parameters.

If we compare the two VOI placing methods, we can see that the elliptical VOIs provide more statistically significant TIs than manual VOIs, and this is especially true in the case of LRR discretization.

## Discussion

In this work, a detailed evaluation of the image voxel discretization procedure's influence on the calculated textural features in T1- and T2-weighted MR studies for three different types of neurological diseases was performed. This is the first study that compares all three discretization methods for MRI and their reliability to the best of our knowledge. Our study revealed that the majority of the selected TIs varied greatly when the discretization parameters (bin width or the number of bins) were changed within a predefined range. In this study, we focused on 18 GLCM, 11 GLSZM, and 11 GLRLM based texture features. Of 12 discretization methods only two (NormInvDiff and Correlation GLCM) were not dependent on the binning parameters for all disease groups and MRI acquisition protocols (an example of this can be

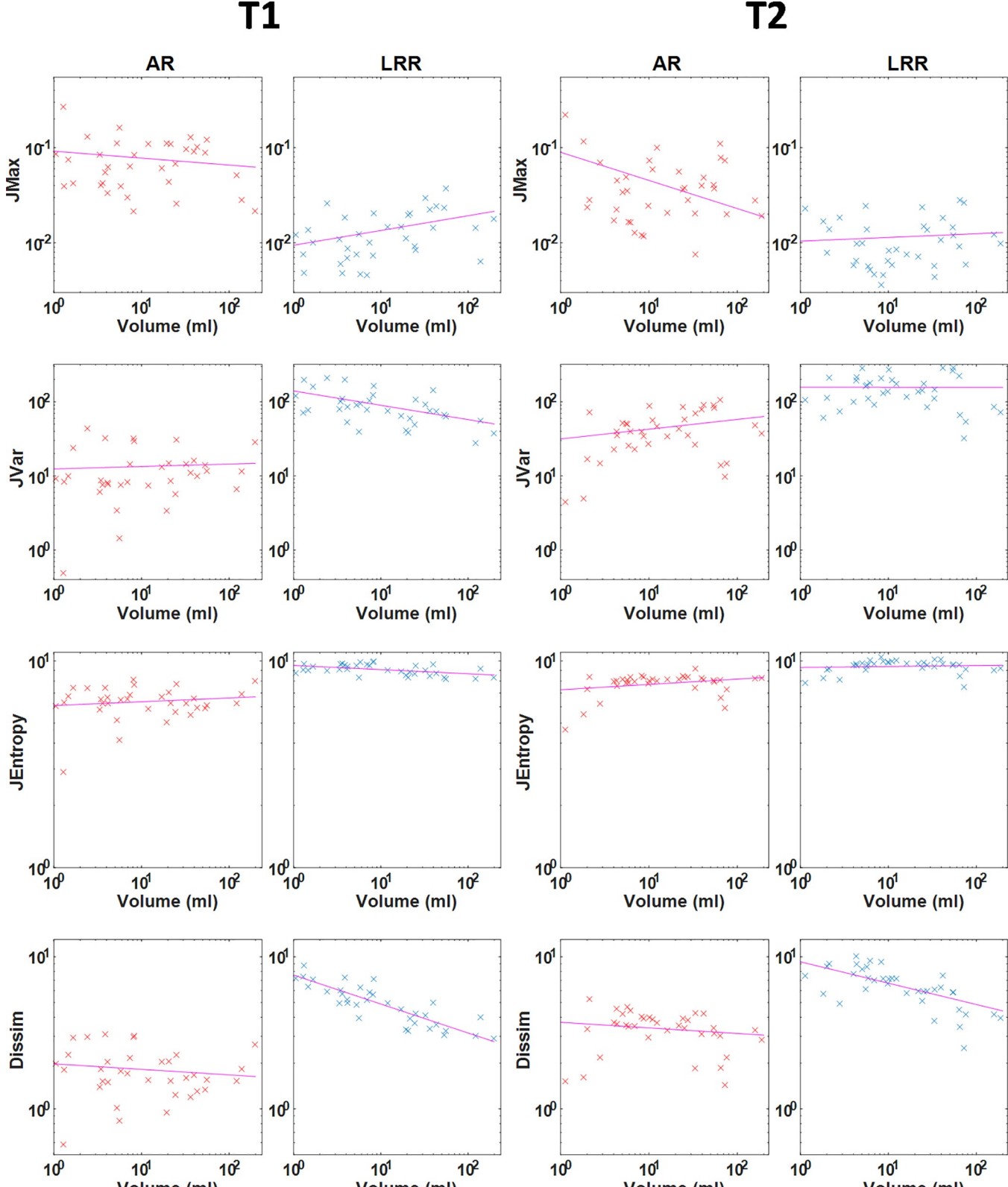

**Fig 5. Pairwise scatter plots of four calculated texture parameters with the lesion volumes including all tumor VOIs employing a bin width of 50 (AR method) and a bin number of 64 (LRR method).** The four TIs shown here are GLCM-based parameters: JMax, Jvar, JEntropy and Dissim. The subplots are

organized in a 4column way, where the two columns on the left and right refer to the T1- and T2-weighted images. The scatter plots have a logarithmic scale on both axes.

seen in Fig 3). While volume independent and acquisition independent parameters are desired, the independence also suggests that their use does not serve as identifiers for disease. In addition, it is important to underline that all the other TIs (N = 38) vary with several orders of magnitude monotonically depending on the binning values. However, the direction of the monotonicity and the span of the values are characteristic for any given TI. These characteristic properties may fundamentally depend on the mathematical expression defining the texture indices, so their analytical examination would greatly assist in examining the applicability of each TI. However, the analyses of the mathematical expressions are yet to be performed. It is often of keen interest to compare TI performance in cohort analyses obtained at different centers and studies. These comparisons, however, are often challenged by the inherent dependency between the TIs performance and the discretization parameters (bin number and sizes). Owing to the dependency between performance and discretization parameters, it is crucial that studies only compare directly to other studies using the same number of bins and discretization techniques. Often times this is not the case; recent studies have been based on a variety of different D values (LRR technique, D = 256, 128, 32) [42–44] and B values (LAR technique, B = 5, 10) [45]. Furthermore, the orders of magnitude do not depend on disease groups (IS, MS, and TU) or imaging sequences analyzed (T1 or T2 image contrast). It is also noticeable, that AR and LRR discretization present the same range span of values for each TI, although the binning values often span several orders of magnitude. The attributes (monotonicity, the span of the TIs) observed for each box plot in Figs 1–3 and S1-S6 Figs in S1 File are concordant with our previous results obtained using PET images [26]. In our previous study on PET data, the

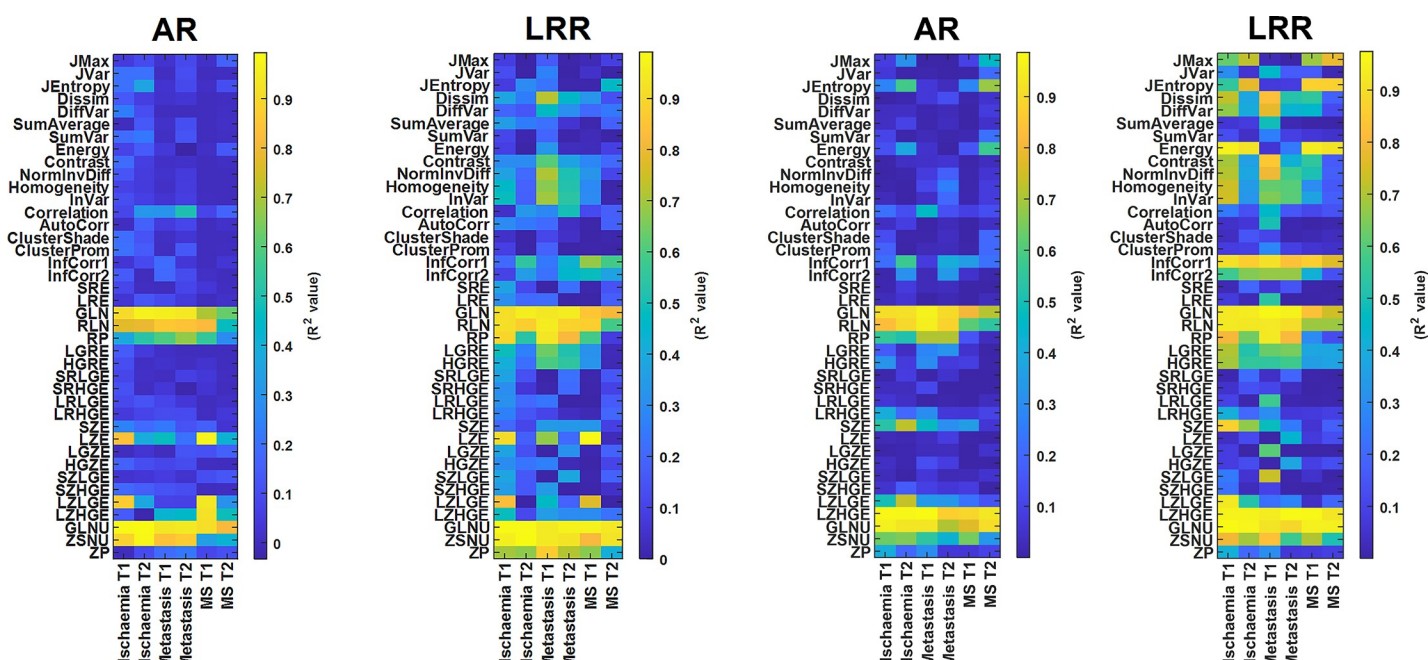

**Fig 6. a.** The $R^2$ values for each correlation analysis between the TIs and the lesion volumes. The predefined bin width (for AR method) and bin number (for LRR method) were 50 and 64, respectively. Data here are obtained using ellipsoid VOIs. **b.** comparison of $R^2$ values for correlation analyses between the TIs and the lesion volumes, after applying manual lesion delineation.

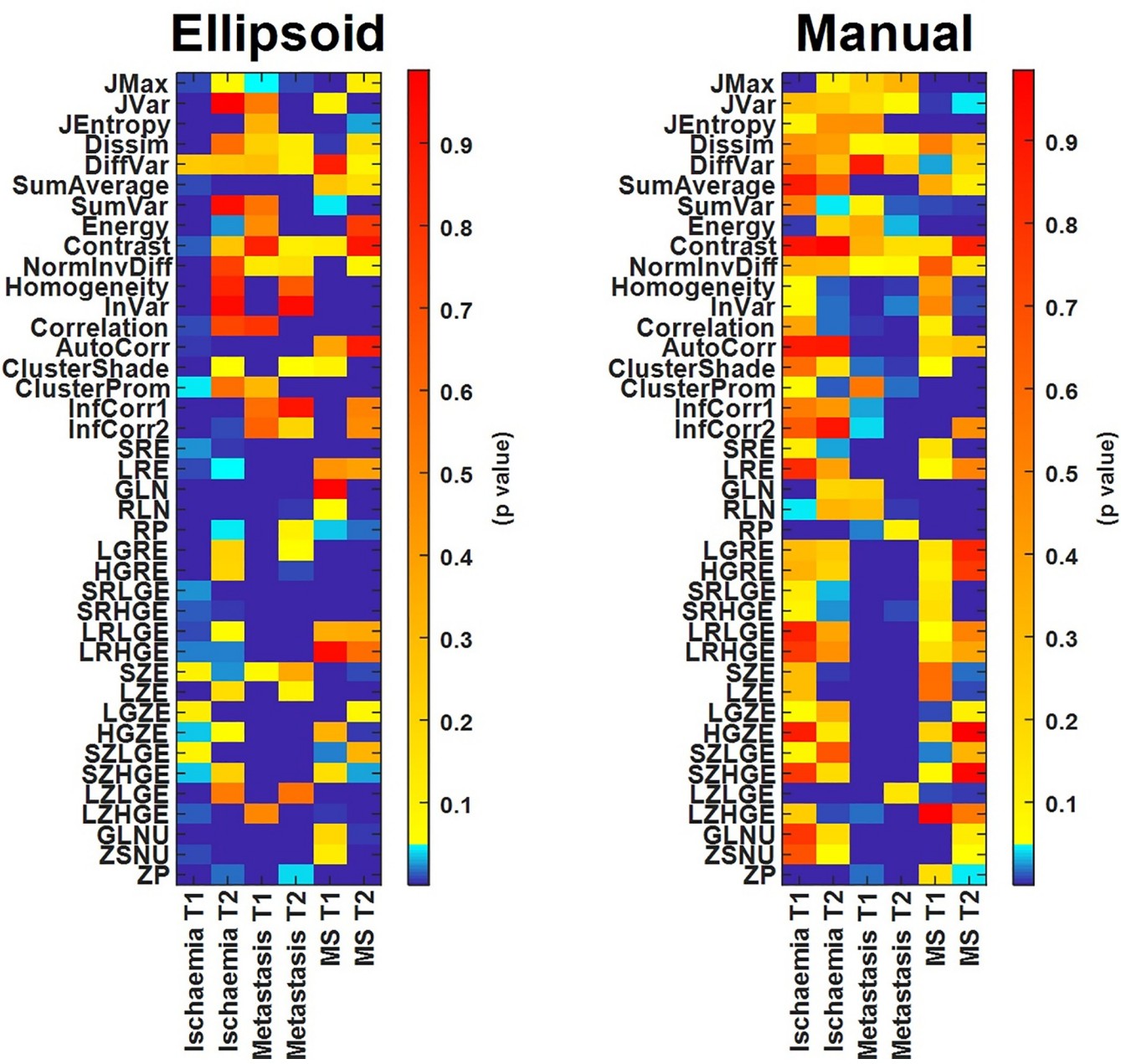

**Fig 7.** Comparison of statistical tests for TIs computed with elliptical VOIs (left panel) and manually drawn VOIs (right panel), applying the LRR discretization method.

AR, LAR, and LRR discretization methods were thoroughly analyzed based on 27 texture parameters. In this context, the same box-plot characteristics are observed for the TIs extracted from our previous PET study and the current study on MRI-based features, although different imaging modalities and diseases were evaluated. Moreover, the variations of the mean values of TIs based on LAR and AR quantification are similar in the current and the previous work. The only one explanation is: that each texture index has a general but highly dependent

# AR

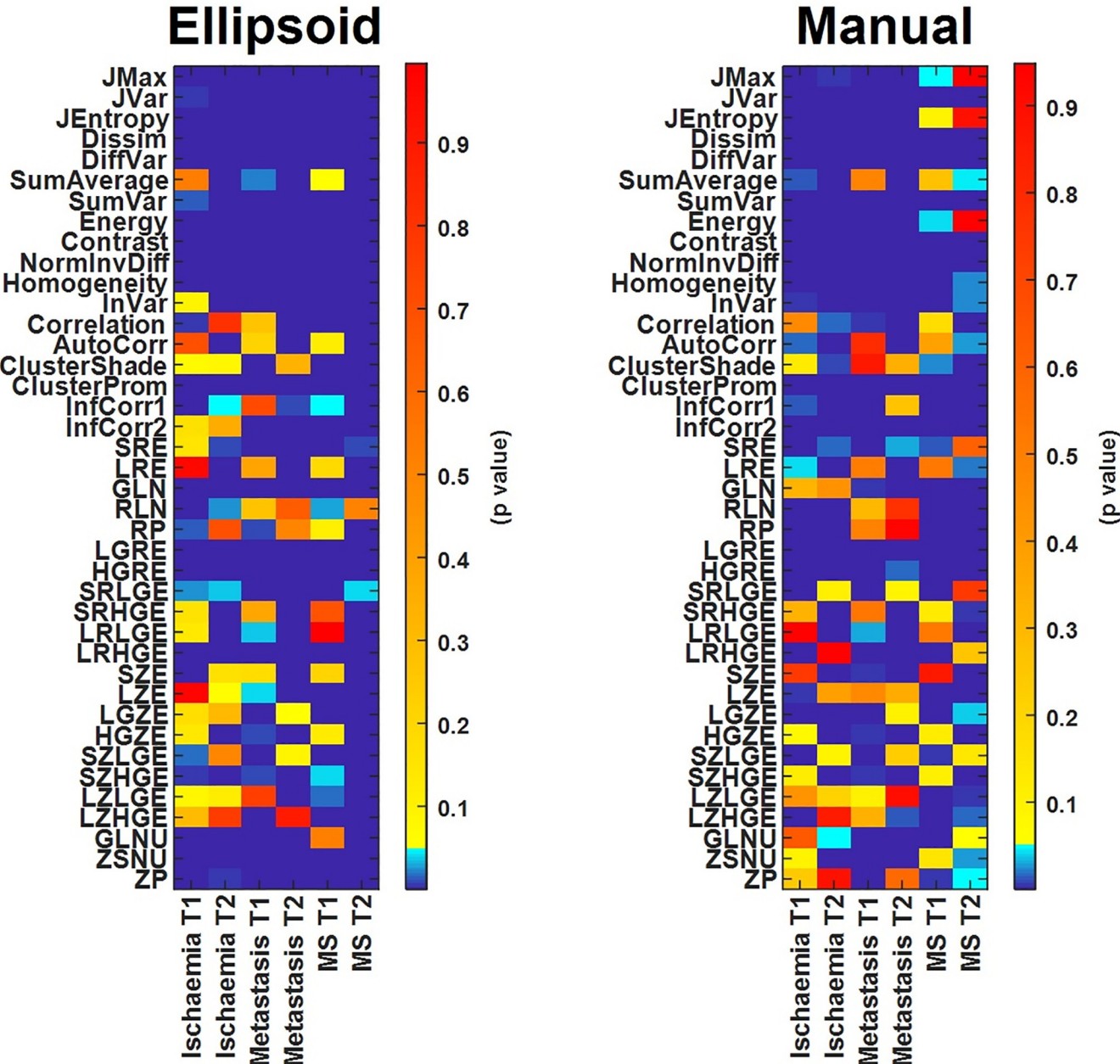

**Fig 8.** Comparison of statistical tests of TIs computed with elliptical VOIs (left panel) and manually drawn VOIs (right panel), applying the AR discretization method.

behavior to the discretization technique and the specific bin parameters. However, the imaging modality, the type of tissue, and the method of segmentation do not have a significant role in this dependence. This finding is in concordance with other studies [29, 31].

The correlation between VOI size and the TIs is another inherent problem relating to performance comparisons of studies [31, 46, 47]. Orlhac et al. proposed the AR resampling

scheme at PET imaging which omits this problem and enables the calculation of TIs without introducing a large bias associated with the tumor volume [32]. In concordance with their study, we analyzed the dependency of the VOI volume and TIs using 426 correlation analyses (Fig 5 and S7-S12 Figs in S2 File). The correlation plots of the TIs obtained for the AR and LRR revealed that most indices are weakly correlated with the tumor volume. Of note, no correlation between the TIs and the disease nor imaging sequence used was observed. For better visualization, we calculated Spearman's correlation coefficients (R) and created a color-coded image representing the R values of all diseases studied with and without the use of MRI contrast. These color-coded images for the cases of ellipsoid and manual segmentation are shown in Fig 6A and 6B, respectively. The $R^2 > 0.5$ ($|R| > 0.71$) cases were considered highly correlated, corresponding to the light green, orange, and even lighter colors. Thus, it is clear from the colormaps that these values are related mainly to the GLRLM and GLSZM groups of the AR and LRR discretization method for both segmentations. Besides the presented parameters, several GLCM parameters are also affected when employing the LRR method and elliptical segmentation, and there will be even more highly correlated GLCM data for LRR discretization and manual segmentation. This effect is unwanted in most cases, as the changes in TIs should depend on the texture, and not the volume. In other words, the LRR discretization is disadvantageous when evaluating the lesion volume—TI correlations, a finding that is in concordance with previous studies [41, 46]. Of note, the $R^2$ value for a given TI does not change substantially as a function of the disease or the T1 or T2 contrast as visualized by the color schemes in Fig 6A and 6B. Some exceptions are RP, LZE, LZLGE, LZHGE, and ZP indices, which belong to both the GLRLM and GLSZM groups. An additional feature of LRR discretization is that the $R^2$ values for the T1 or the T2 contrast in each disease group are slightly but characteristically different for most TIs.

Figs 7 and 8 show the color-coded p-values for the LRR and AR discretizations from the hypothesis tests for among all disease and control areas. Both figures clearly show the different effects of the two distinct segmentations. The assumed hypothesis was that TI values from the healthy areas differ from those obtained from the disease areas. An increased number of suitable texture features (cold color areas, p-value < 0.05) was observed when employing the AR technique instead of the LRR in both delineation methods. In addition, the manual segmentation gave fewer significantly different TIs, and this was even more pronounced for LRR discretization. A possible explanation for this phenomenon might be that by applying manual segmentation, tissues outside of the lesion affect the calculation of TIs, compared to the case of elliptical VOIs where only tissues inside the lesions get selected. Based on AR discretization and both segmentations (Fig 8) the following TIs are the most promising in MR images: Jvar, Dissim, SumVar, Contrast, NormInvDiff, ClusterProm, LGRE. Another interesting finding is that the TIs that correlated with the volume as shown in Fig 6A and 6B have no noticeable interrelationship with how well they can be used in the hypothesis test.

In general, only a few studies in the literature have evaluated the effect of the grey level discretization methods for MRI acquisitions. Molina *et al.* (2016) analyzed how the preselected number of bins for the LRR method changes the texture features when using sixteen heterogeneity measures. In their study, the main conclusion was that none of the considered TIs were robust with changing bin parameters. Goya-Outi *et al.* (2018) evaluated the impact of the AR and LRR methods in a study including 30 patients with diffuse intrinsic pontine gliomas, using 4 MR contrasts/sequences. They showed that the textural measures obtained with the AR discretization were more consistent with the visual assessments than the same comparison of the visual assessment and the LRR method. Further studies also support that the IBSI recommended MRI related LRR discretization should be changed to the absolute method (AR or LAR) [29, 31]. Recently, Carré et al. also investigated the impact of the discretization methods

on radiomics features with T1- and T2-weighted MRI images [33]. They found that the selection of optimal D or B is not trivial. Their results showed that a high D value might increase feature robustness for most radiomics parameters, but interestingly their classification performance decreased in the case of the T2-flair sequence. This was in good agreement with our observations (Figs 7 and 8).

The superior soft tissue contrast of MR images renders the automatic segmentation of lesions and pathological tissues a challenge. Consequently, labor-intensive manual delineations are often employed for texture analyses. However, several semi-automatic or automatic algorithms are available which might be beneficial for future studies [31]. In general, the semi-automatic and automatic algorithms employ different approaches that aid the segmentation of 3D volumes when evaluating MR images. Besides the more advanced lesion delineation method, the use of squares or circles is a possible alternative [41, 48]. The resulted segmentation is influenced by several factors: partial volume effect, intensity inhomogeneity, artifacts, proximity and overlap of the gray and white level pixels. In this study, we compared the effects of two basic segmentation strategies, the manual and the 3D elliptical delineation. To compare healthy tissue and tissue with disease, two VOIs were used. One VOI was inserted within the pathological area, while a volume matched VOI was inserted in healthy tissue on the co-lateral side. Our analyses show that fewer statistically significant texture parameters could be obtained when manual VOI segmentation was applied. This can be explained by the fact that the voxels in the border area, due to the inadequate spatial resolution, might contain a mixture of the surrounding tissue and the tissue from the given lesion. Accordingly, the calculated texture indices will also be distorted. The 3D ellipsoid segmentation method yielded the most significantly different TIs for any discretization, these are illustrated in Figs 7 and 8.

We report three main limitations to this study. First, we did not analyze how the different image normalization strategies would affect our results. Several normalization techniques have been proposed [31, 49, 50], and at this point, it is uncertain which proposed method is optimal. Secondly, the spatial resolution parameter is also a determinant factor in the texture analysis [26, 51] that we did not include its evaluation in our study. However, the scope of this study and the evaluations required for it are already beyond the scope of this study. Last, we focused only on the group of GLCM, GLSZM, and GLRLM based TIs without considering other features already published to highlight the impact of the three discretization methods evaluated. We are planning to further extend our study to analyze the effect of image normalization and automatic lesion segmentation on the image discretization step.

## Conclusions

We compared the effects of three discretization methods in brain MRI images of three different types of diseases. We found that the values of TIs characteristically depend on the applied binning parameters, making the choice of these parameters non-trivial, therefore it is critical that comparative studies only compare directly to other studies using the same number of bins and discretization techniques. However, the characteristic dependence does not change with the type of disease, nor the applied MR-sequence. In general, it can even be stated that the same discretization dependence is observed for MRI and PET modality. The AR and LAR methods provide more significantly different TI values than LRR, when comparing calculations on control and pathological brain areas. We also found that in general the TIs weakly correlate with the volume of lesions, but several GLCM based texture parameters showed a higher correlation when the LRR discretization method was selected. Therefore, we recommend using AR or LAR discretization instead of LRR in the case of brain MRI images. Applying manual segmentation instead of elliptical VOIs, fewer statistically significantly different

TIs can be obtained between control and diseased areas, and the level of correlation between TIs and volume of lesions tend to increase. In the texture analysis of MR studies, studies and publications should report on all important parameters and methods related to data collection, corrections, normalization, discretization, and segmentation.

## Supporting information

**S1 File. Box-and-whisker plot and hypothesis test results for all radiomics features.** (XLSX)

**S2 File. Data and results of the correlation analysis.** (XLSX)

## Author Contributions

**Conceptualization:** Gergő Veres, Ervin Berényi, László Balkay.

**Data curation:** Aron K. Krizsan.

**Formal analysis:** Norman Félix Vas, Monika Béresová, Attila Forgács.

**Funding acquisition:** László Balkay.

**Investigation:** Gergő Veres, Monika Béresová.

**Methodology:** Martin Lyngby Lassen, Aron K. Krizsan, Attila Forgács.

**Project administration:** László Balkay.

**Resources:** Ervin Berényi.

**Software:** Norman Félix Vas.

**Supervision:** László Balkay.

**Validation:** Martin Lyngby Lassen, Aron K. Krizsan.

**Visualization:** Gergő Veres, Norman Félix Vas.

**Writing – original draft:** Gergő Veres, Norman Félix Vas, László Balkay.

**Writing – review & editing:** Gergő Veres, Norman Félix Vas, Martin Lyngby Lassen, Monika Béresová, Aron K. Krizsan, Attila Forgács, Ervin Berényi, László Balkay.

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
