## [Decision Letter · Decision Letter 0]

13 Apr 2021

PONE-D-21-08818

Effect of grey-level discretization on texture feature on different weighted MRI images of diverse disease groups

PLOS ONE

Dear Dr. Veres,

Thank you for submitting your manuscript to PLOS ONE. After careful consideration, we feel that it has merit but does not fully meet PLOS ONE’s publication criteria as it currently stands. Therefore, we invite you to submit a revised version of the manuscript that addresses the points raised during the review process.

Please ensure English language improvements as indicated.

Please consider incorporating Reviewer 2`s proposals to the manuscript, too.

We look forward to receiving your revised manuscript.

Kind regards,

Domokos Máthé

Academic Editor

PLOS ONE

Journal Requirements:

2. In ethics statement in the manuscript and in the online submission form, please provide additional information about the patient records used in your retrospective study. Specifically, please ensure that you have discussed whether all data were fully anonymized before you accessed them and/or whether the IRB or ethics committee waived the requirement for informed consent. If patients provided informed written consent to have data from their medical records used in research, please include this information.

'The funders had no role in study design, data collection and analysis, decision to

publish, or preparation of the manuscript.'

Additional Editor Comments (if provided):

Reviewers' comments:

Reviewer's Responses to Questions

**Comments to the Author**

1. Is the manuscript technically sound, and do the data support the conclusions?

Reviewer #1: Yes

Reviewer #2: Yes

2. Has the statistical analysis been performed appropriately and rigorously? 

Reviewer #1: Yes

Reviewer #2: Yes

3. Have the authors made all data underlying the findings in their manuscript fully available?

Reviewer #1: Yes

Reviewer #2: Yes

4. Is the manuscript presented in an intelligible fashion and written in standard English?

Reviewer #1: Yes

Reviewer #2: Yes

5. Review Comments to the Author

Reviewer #1: The article basically examines the normalisation of radiological methods that are very popular today. Lots of parameters were evaluated and classified based on accepted statistical and radiology methods. The article is well written and after a minor revision, I suggest accepting. My smaller remarks:

1) The article contains spellings and grammatical errors in many places. Please run through and fix these.

2) In the results section, please describe which statistical methods were used to obtain the described results. Indeed, page 6 contains these methods.

3) I suggest inserting a new figure which presents the effect of the discretisation processes on images.

4) Please try to discuss why you recommend using AR or LAR discretization instead of LRR. The conclusion is very diffuse and questionable.

Reviewer #2: An already thorough analysis.

Minor things that probably not required for publication, but would make the article much better, but including them would probably grow the article and the study way out of proportion, and probably can be addressed in a more encompassing future article:

To normalization, discretization mentioned in the article, I would add spatial resolution as well as parameters governing feature extractions. All these scaling factors could affect feature statistics. Examining only one aspect of these is concern raising enough. Analysing the effect of all with respect to various diseases require a large dataset and combinatorically more complicated work, therefore I do not expect the authors to embark broadining the study.

Still a little bit more emphasis on the mathematical reasons behind discretization causing some applied features to monotonically change while others not. The numerical assessment is very thorough, but understanding the math should help reasoning for the importance of certain aspects.

In addition, it is not entirely clear that this article has optimal recommendations for all or any of these parameters. Therefore this study can be regarded rather as an examplary attention raising article. As such, it should communicate this aspect more clearly. Thus I would like to see more highlighting in the abstract/intro/discussion this recommendation for the wider research community: studies and publications should be reporting discretization, normalization, resolution, feature parameters and methods.

6. PLOS authors have the option to publish the peer review history of their article (what does this mean?). If published, this will include your full peer review and any attached files.

Reviewer #1: No

Reviewer #2: No

---

## [Author Response · Author response to Decision Letter 0]

19 May 2021

Reviewer #1: The article basically examines the normalisation of radiological methods that are very popular today. Lots of parameters were evaluated and classified based on accepted statistical and radiology methods. The article is well written and after a minor revision, I suggest accepting. My smaller remarks:

1) The article contains spellings and grammatical errors in many places. Please run through and fix these.

We made rigorous grammar correction throughout the text.

2) In the results section, please describe which statistical methods were used to obtain the described results. Indeed, page 6 contains these methods.

For correlation analysis and the related plots the Spearman’s correlation coefficients were calculated, and in the case of hypothesis tests we performed Wilcoxon ranksum tests among all disease and control areas. These details have been added to the result section (page 11).

3) I suggest inserting a new figure which presents the effect of the discretisation processes on images.

A representative flow chart was inserted in the Methods section (Fig 1b). In this figure a T1-weighted MR image comprises a larger tumor, and after the segmentation the effect of the three different discretizations are presented. The AR and LAR discretizations smooth and equalize the contrast much better within the segment, which is most easily observed at lower values (blue colors) of the images.

4) Please try to discuss why you recommend using AR or LAR discretization instead of LRR. The conclusion is very diffuse and questionable. 

There are basically two attributes of what we think is important. First, the AR and LAR methods provide more significantly different TI values than LRR, when comparing the calculated TIs on control and pathological brain areas. And second, we found that in general the TIs weakly correlate with the volume of lesions, but several GLCM based texture parameters showed a higher correlation when the LRR discretization method was selected. Furthermore, since smallest correlation is more advantageous for proper texture analysis, the LRR method is disadvantageous in this respect as well.

 

Reviewer #2: An already thorough analysis.

Minor things that probably not required for publication, but would make the article much better, but including them would probably grow the article and the study way out of proportion, and probably can be addressed in a more encompassing future article:

To normalization, discretization mentioned in the article, I would add spatial resolution as well as parameters governing feature extractions. All these scaling factors could affect feature statistics. Examining only one aspect of these is concern raising enough. Analysing the effect of all with respect to various diseases require a large dataset and combinatorically more complicated work, therefore I do not expect the authors to embark broadining the study. 

We agree with the reviewer that the image spatial resolution is another important parameter that need to be considered in a radiomics analysis. Recently, Zwanenburg published an article concluding that differences in voxel size substantially affects the measurement agreement in PET and CT investigations, thus pixel size harmonization is recommended, if multicentric study need to be accomplished (https://doi.org/10.1007/s00259-019-04391-8). We carefully set the spatial resolution for each patient scan and both contrast (T1 and T2), concordance with our standard clinical MRI protocols. The extension of the current investigation, including an analysis of the spatial resolution, cannot easily be achieved because the manuscript already contains a large number of data and figures related. We plan the proposed analysis in the near future. 

Still a little bit more emphasis on the mathematical reasons behind discretization causing some applied features to monotonically change while others not. The numerical assessment is very thorough, but understanding the math should help reasoning for the importance of certain aspects. 

We thank the suggestion for the reviewer. These characteristic properties (type and degree of monotonicity) may fundamentally depend on the mathematical expression defining the texture indices, so their analytical examination would greatly assist in examining the applicability of each TI. However, the analyses of the mathematical expressions are yet to be performed. We agree that this mathematical analysis would be advantageous for the detailed understanding of each TI, and we intend to perform this investigation in a next study. The discussion section was updated according in manuscript.

In addition, it is not entirely clear that this article has optimal recommendations for all or any of these parameters. Therefore this study can be regarded rather as an examplary attention raising article. As such, it should communicate this aspect more clearly. Thus I would like to see more highlighting in the abstract/intro/discussion this recommendation for the wider research community: studies and publications should be reporting discretization, normalization, resolution, feature parameters and methods. 

We agree with the reviewer, the main recommendations were highlighted in the abstract, discussion, and conclusion parts.

---

## [Editor Report · Decision Letter 1]

7 Jun 2021

Effect of grey-level discretization on texture feature on different weighted MRI images of diverse disease groups

PONE-D-21-08818R1

Dear Dr. Veres,

We’re pleased to inform you that your manuscript has been judged scientifically suitable for publication and will be formally accepted for publication once it meets all outstanding technical requirements.

Kind regards,

Domokos Máthé

Academic Editor

PLOS ONE
---

## [Editor Report · Acceptance letter]

9 Jun 2021

PONE-D-21-08818R1 

Effect of grey-level discretization on texture feature on different weighted MRI images of diverse disease groups  

Dear Dr. Veres:

I'm pleased to inform you that your manuscript has been deemed suitable for publication in PLOS ONE. Congratulations! Your manuscript is now with our production department. 

Kind regards, 

on behalf of

Dr. Domokos Máthé 

Academic Editor

PLOS ONE